# Unsupervised Exercise Training Was Not Found to Improve the Metabolic Health or Phenotype over a 6-Month Dietary Intervention: A Randomised Controlled Trial with an Embedded Economic Analysis

**DOI:** 10.3390/ijerph18158004

**Published:** 2021-07-28

**Authors:** Wendy Hens, Dirk Vissers, Nick Verhaeghe, Jan Gielen, Luc Van Gaal, Jan Taeymans

**Affiliations:** 1Faculty of Medicine and Health Sciences, University of Antwerp, 2610 Antwerp, Belgium; Dirk.vissers@uantwerpen.be; 2Department of Cardiology, Antwerp University Hospital, 2650 Edegem, Belgium; 3Faculty of Medicine and Health Sciences, Ghent University, 9000 Ghent, Belgium; Nick.Verhaeghe@UGent.be; 4Department of Radiology, Antwerp University Hospital, 2650 Edegem, Belgium; jan.gielen@uza.be; 5Department of Endocrinology, Antwerp University Hospital, 2650 Edegem, Belgium; luc.vangaal@uza.be; 6Health Department, Bern University of Applied Sciences, 3008 Bern, Switzerland; jan.taeymans@bfh.ch; 7Faculty of Sport and Rehabilitation Sciences, Vrije Universiteit Brussel, 1050 Elsene, Belgium

**Keywords:** exercise, overweight, body fat distribution, ectopic fat

## Abstract

Ectopic fat leads to metabolic health problems. This research aimed to assess the effectiveness of a hypocaloric diet intervention together with an unsupervised exercise training program in comparison with a hypocaloric diet alone to reduce ectopic fat deposition. Sixty-one premenopausal women with overweight or obesity participated in this controlled trial and were each randomised into either a usual care group (hypocaloric diet) or intervention group (hypocaloric diet + unsupervised exercise training). Ectopic fat deposition, metabolic parameters, incremental costs from a societal perspective and incremental quality-adjusted life years (QALYs) were assessed before, during and after the six-month intervention period. In the total sample, there was a significant decrease in visceral adipose tissue (VAT: −18.88 cm^2^, 95% CI −11.82 to −25.95), subcutaneous abdominal adipose tissue (SAT: −46.74 cm^2^, 95% CI −29.76 to −63.18), epicardial fat (ECF: −14.50 cm^3^, 95% CI −10.9 to −18.98) and intrahepatic lipid content (IHL: −3.53%, 95% CI −1.72 to −5.32). Consequently, an “adapted” economic analysis revealed a non-significant decrease in costs and an increase in QALYs after the intervention. No significant differences were found between groups. A multidisciplinary lifestyle approach seems successful in reducing ectopic fat deposition and improving the metabolic risk profile in women with overweight and obesity. The addition of unsupervised exercise training did not further improve the metabolic health or phenotype over the six months.

## 1. Introduction

Weight loss and weight loss maintenance is the core objective when aiding people with overweight and obesity. This goal can be reached using different strategies, such as non-invasive forms of behaviour therapy, pharmacotherapy or invasive approaches, such as weight loss surgery. Although a comprehensive lifestyle change approach with a reduced caloric diet and increased physical activity to achieve weight reduction is supported in obesity guidelines, this combined therapy is not always prescribed. Dieting is considered the cornerstone of obesity management. The media’s messages about the futility of exercise have led to confusion about the usefulness of exercise training. It is often seen that the role of exercise training is not emphasised. Frequently, a more general public health approach of physical activity promotion (“You should exercise more regularly”) is used instead of an individualised exercise prescription [1]. An exercise prescription commonly refers to a specific plan of sport-related activities that are designed for a specific purpose. Basic components of exercise prescription include recommendations for optimal exercise frequency, intensity, duration, mode and progression based on individual patient needs, preferences and limitations in order to achieve certain goals. These sport-related activities can be supervised (in groups or individually) or unsupervised with limited or extensive feedback.

Exercise training does not contribute to initial weight reduction when combined with a hypocaloric diet. However, exercise training has the potential to optimise body composition, enhance cardiovascular fitness, consolidate achieved weight loss and reduce metabolic risk [2].

Obesity is associated with weight-related comorbidities and an increased risk of morbidity and mortality [3]. A cost-of-illness analysis revealed the same pattern in metabolically healthy women with overweight and obesity [4]. This population shows higher overall and overweight-related comorbidities, which result in higher (healthcare) costs and lower average health utility as compared to the general population. Although this presents a major public health challenge, individuals with overweight or obesity should be encouraged to optimise their weight. In the absence of a significant weight loss, exercise training has been shown to reduce the presence of weight-related co-morbidities and further slow the development and progression of cardiometabolic disease [5]. The risk reduction may be related to increased muscle volume and a reduction in adipose tissue. Adipose tissue is known to be an active endocrine and paracrine organ that influences local and general insulin resistance, inflammation, lipid levels and, hereby, metabolic health [6]. Spillover of adipose tissue will be located in organs or in places that do not have the functionality for fat storage, for instance, in and around the liver, the heart, the skeletal muscles or abdominal intestines, which is called ectopic fat deposition [7].

Policymakers are facing a problem on how to set priorities in the allocation of limited healthcare resources to medical or public health interventions. Knowledge regarding these issues can be obtained by performing health economic evaluations of treatments, providing payers and governments with better insights into how to spend the available resources most efficiently (i.e., in a cost-effective way). Research is needed to provide the scientific, economic and patient-oriented rationale regarding the impact of exercise prescription on metabolic risk in people with overweight or obesity.

Therefore, the research question of this randomised controlled trial (RCT) was:

What is the effectiveness of a hypocaloric diet intervention alone or in combination with a free of charge but unsupervised exercise training program on the reduction of ectopic fat deposition and amelioration of the metabolic risk in women with overweight and obesity? Additionally, general costs were inventoried to take cost effectiveness of the interventions into account.

## 2. Materials and Methods

The Consolidated Standards of Reporting Trials (CONSORT) guidelines 2010 and the Consolidated Health Economic Evaluation Reporting Standards (CHEERS) statement were used to report this single-blinded RCT [8,9]. The study protocol was described extensively in a previous publication [10].

Participants: Women with overweight and obesity were recruited by endocrinologists at the obesity clinic of the Antwerp University Hospital (tertiary referral facility). Additionally, poster recruitment was used at the University of Antwerp and the Antwerp University Hospital. In brief, premenopausal women without cardiovascular consequences (e.g., diabetes type 2, thyroid problems, or cardiac diseases) with a stable body mass index (BMI) of at least 27 kg/m^2^ were included. Participants who took medications that influenced body weight or metabolism (e.g., tricyclic antidepressants) were excluded. Since medical imaging techniques were used to quantify the ectopic fat deposition, general exclusion criteria regarding computed tomography (CT) and magnetic resonance imaging (MRI) were applicable.

Intervention: In this six-month intervention study, each participant was either assigned to a usual care group or an intervention group. To reduce the imbalance between baseline values in each group, a minimisation technique was used based on age, the occurrence of symptoms of the metabolic syndrome and the amount of visceral adipose tissue (VAT).

Usual care group (UC group): A hypocaloric diet with an energy deficit of 500 kcal/day based on each individual’s resting metabolic rate was prescribed. Subjects participated in a total of seven individualised dietary counseling sessions. These consultations had a duration between 30 and 60 min and were used to introduce the principles of balanced meals, energy restriction and portion sizes and to discuss dietary patterns and recommend specific changes. At each visit, nutritional compliance was estimated by the dietician on a rating numeric scale of 0 to 10 based on participant’s feedback without taking possible weight loss variability into account. Subjects were labeled as non-adherent when the mean adherence score was lower than 6/10. The participants of the UC group were asked to continue with their normal physical activity pattern. Participants received motivational support by mail or telephone twice per month, following a fixed protocol.

Intervention group (I group): A hypocaloric diet was supplemented with a free-of-charge, individualised training program in a fitness or health centre near the participant’s home. The prescribed training was a combination of aerobic and strength training that was performed three times per week. The exercise sessions were non-supervised but the exercise regimen was instructed by the research physiotherapist on location. It was based on exercise physiological principles, personal physical inconveniences and possible individual barriers to physical activity and exercise [11]. The aerobic exercise regimen was standardised by applying the same exercise volume and progression in each participant based on the results of a maximal cardiopulmonary exercise test (CPET; electronic bicycle ergometer: LODE–Corival; gas analyser system: JAEGER–Oxycon Pro). The intensity of the aerobic exercise training was heart rate-driven, which was set to 90–95% of the heart rate achieved at the respiratory compensation point. Aerobic training was performed on three to four different cardio devices for a total of 30 to 45 min. On each training day, aerobic training was completed in combination with core stability training and four strength exercises on isotonic strength training devices for a total of 20 min. After 6 months, the compliance was measured on a scale from 0 to 9 depending on session duration, exercise intensity and weekly training frequency. Subjects were labeled as non-adherent when the mean compliance score was lower than 6/9. Participants received motivational support by mail or telephone twice per month following a fixed protocol.

Primary outcome measures: The primary clinical outcome measures were the changes in ectopic fat deposition from baseline to three months and six months, measured at different locations.

A single-slice CT of the abdomen (GE, Lightspeed VCT) was used to evaluate the VAT (cm^2^) and subcutaneous abdominal adipose tissue at the L4–L5 region according to previously described methods [12]. Multi-slice end-diastolic ECG-triggered CT was used to measure pericardial fat areas (PCF, cm^3^) and epicardial fat areas (ECF, cm^3^). PCF and ECF areas were manually delineated slice-by-slice, starting at the pulmonary artery and ending at the apex of the heart using the workstation (GE, AW Volumeshare 2) according to previously described methods [13]. Tissue with attenuation values in the interval of −30 to −190 Hounsfield units was considered fat [14]. 1H magnetic resonance spectroscopy (1H-MRS, SIEMENS MAGNETOM Prisma 3T) was used to quantify ectopic fat deposition in the right tibialis anterior muscle (Intra MyoCellular Lipids or IMCL/creatine) and the liver (intrahepatic lipid content or IHL% relative to water). Pre-processing of all spectra was standardised using the jMRUI2XML software and quantification was done using Amares in the jMRUI software version 5.2 [15,16]. The instructions given were to avoid high-fat-loaded foods and doing excessive physical effort three days before the medical imaging techniques.

Secondary outcome measures: Anthropometric measurements were performed in fasting conditions. Waist circumference (WC) was measured at the midlevel between the lower rib margin and the iliac crest. Body composition was determined using bio-impedance analysis (BIA), as described by Lukaski et al. [12], and body fat mass percentage was calculated using the formula of Sun et al. [17]. Systolic and diastolic blood pressure (SBP and DBP) was determined using a mercury sphygmomanometer at the right arm after lying down for at least five minutes. Blood samples (from an antecubital vein) were taken at the start of the study and after six months and were analysed for fasting glucose (FG) and the lipid profile (total cholesterol, high-density lipoprotein (HDL-C), triglycerides (TG)).

Costs and quality of life measurements were assessed at baseline and after three and six months using self-reported cost diaries and the EuroQol (EQ-5D-5L) questionnaires [4,18]. In the cost diary, units consumed over the last three months, as well as prices for medication and visits to health providers, such as physicians, specialists, nurses, dieticians and physiotherapists, were asked for. Furthermore, consumption and the prices of hospitalisation, surgery and medical examinations were assessed over the same period, together with the number of days absent from work due to any health problem. The EQ-5D-5L algorithm with the UK–Tariff was used to transform the individual health profiles into utility indices (i.e., health-related quality of life weight ranging between 0 and 1 with “1” indicating perfect health and “0” dead). Direct medical costs (e.g., medication use, (para)medical consultations and assessments, hospitalisation), direct non-medical costs (e.g., braces, incontinence pads) and indirect non-medical costs (absenteeism from work, unemployment) were calculated. Costs of healthcare consumption were calculated using the following formula: units consumed × unit price, and were reported in a non-aggregated form and expressed in EUR. No discounting was needed given the time horizon of six months. If participants did not mention the unit prices of care providers, the fees were searched for on the website of the National Institute for Health and Disability Insurance (NIHDI) [19,20]. Hereby, it was assumed that patients consulted accredited healthcare workers and the patient had the right to the normal reimbursement rate. If it was not possible to reconstruct the true healthcare cost, the billing service tariff of the University Hospital of Antwerp was applied. Medication costs were calculated based on the units consumed through the Belgian commented online drug compendium. Absenteeism was analysed as an indirect cost and valued as 288.00 EUR per day following the whitepaper of SECUREX, which is a company providing social secretary services in Flanders [21]. The cost of unemployment was estimated to be 92.62 EUR per day, based on the report from the European Federation for Services to Individuals [22]. This study was conducted from a societal perspective. Quality-adjusted life years (QALYs) were calculated by multiplying the utility index level for a given condition by the period an individual lived with the condition. Differences in costs and QALYs between the UC and I groups allowed us to calculate the incremental cost-effectiveness ratio (ICER) and hereby define the cost-effectiveness of the intervention compared with the control group. The ICER was calculated as follows: ICER = (Cost_I_ − Cost_UC_)/(QALY_I_ − QALY_UC_). In the case of no significant between-group differences found in the primary and secondary outcomes, the average cost–utility ratios (ACUR) were obtained by dividing the societal direct medical, direct non-medical and indirect costs by the utility score for each group (I and UC) separately. ACURs were reported as EUR per QALY. Changes in costs and QALY between baseline and six months were calculated. Although the fact that the dietary and exercise treatments were free of charge for participants in this study, the real cost was estimated by questioning travel costs and extra costs related to the treatment (e.g., sporting gear). This was also added to the mean cost of seven dietary counseling sessions and the mean cost of a fitness membership over six months.

Data analysis: Assuming a two-sided α of 0.05 and a power of 0.95, a sample size of 39 women was required to detect IHL changes. The effect size was calculated based on IHL data from the exercise group in the Hallsworth et al. study [23]. Since the dropout rate in lifestyle intervention studies can be up to 35% in people with overweight, it was concluded that 60 women should be included at baseline [24,25]. A repeated-measures ANOVA was used to detect changes in the total group over three time points (T0, T3 and T6 months) and a paired samples *t*-test was used to detect changes in costs and QALYs (measured at baseline and after six months). Sphericity was checked with Mauchly’s test of sphericity. When sphericity was violated, a Greenhouse–Geisser correction was used. Relationships between changes in ectopic fat deposition and changes in other metabolic parameters were assessed. All values are expressed as means ± standard deviations (SDs). Linear mixed models were fitted to test whether the change in ectopic fat deposition over the three time points was different between the UC and I groups. For each ectopic fat parameter, a separate model was fitted, with ectopic fat deposition as the dependent (outcome) variable. The independent variables (fixed effects) included time, treatment and interaction. To account for the non-independence between observations within the same individual, a random intercept term for an individual was added to the model. The significance of the interaction term, which tested whether the effect of the UC group was different from the I group, was assessed by comparing the model with the group × time interaction term to a model that only included the main effects of time and treatment using an F-test with a Kenward–Roger correction for degrees of freedom. Furthermore, it was tested whether there was a difference in outcome (fat) between the three time points. Pairwise differences between time points were tested using Tukey’s correction for multiple comparisons.

## 3. Results

### 3.1. Flow of Participants

The study participants’ flow chart is shown in Figure 1.

The dropout rate was 15%. There were five “early dropouts” (before measurements at the third month) and four “late dropouts” (before measurements at month six). Reasons for leaving the study were similar in both groups and were mostly due to a lack of time. There was no significant difference in baseline values between the UC and I groups regarding primary and secondary outcomes (Table 1). NAFLD and metabolic syndrome were seen in seventeen (28%) and three (5%) participants in the total sample, respectively.

The UC and I groups were comparable at baseline. Descriptive data of the study participants can be found in Table 1.

### 3.2. Changes in Ectopic Fat Deposition and Metabolic Risk Factors in the Total Sample (UC + I Groups)

VAT and SAT significantly decreased over the intervention period of six months (F(2.92) = 18.190, *p* < 0.001; resp. F(2.92) = 36.362, *p* < 0.001). Furthermore, ECF (F(2.94) = 32.736, *p* < 0.001) and intrahepatic lipid content (F(2.96) = 9098, *p* < 0.001) decreased significantly after six months in the total sample. PCF and IMCL did not change significantly. Figure 2 shows the changes in the total sample at ectopic fat deposition sites.

WC in the total group changed significantly after six months (*p* < 0.001). Moreover, TG decreased significantly after six months (*p* = 0.003). No significant changes were found for HDL-C (*p* = 0.664), FG (*p* = 0.633), DBP (*p* = 0.605) and SBP (*p* = 0.308). Figure 3 shows the changes in the metabolic parameters in the total sample.

Few correlations were found for the total sample between the changes in metabolic parameters and changes in the ectopic fat deposition: SBP was the only variable that correlated with the change in VAT after three months (R^2^ = 0.427, *p* = 0.003). Furthermore, changes in WC were weakly to moderately and positively correlated with VAT changes (*p* < 0.05).

### 3.3. Differences between the UC and I Groups

No significant differences were found in the changes in mean ectopic fat deposition or mean metabolic parameters over the six months between the two groups. In Appendix A, the clinical changes over time in each group are presented.

In addition, changes and 95% confidence intervals of the changes in the primary outcome are presented in Figure 4. A significant reduction in ectopic fat deposition was found after three months of intervention in both groups, except for IMCL and IHL, and PCF in the I group. Ectopic fat reductions were sustained after six months of intervention, except for PCF in both groups.

Excluding non-adherent participants in a sensitivity analysis revealed no new results.

### 3.4. Cost-Effectiveness

One participant in the UC group was excluded from this analysis since she had outlier values regarding direct medical and non-medical costs. This woman showed direct medical and non-medical costs that were 10-fold the mean cost in the UC group. Additionally, the indirect costs of productivity loss, such as unemployment and absenteeism, were major cost drivers in this study. Since the baseline costs of unemployment and absenteeism were twice as high in the UC group in comparison with the I group, these expenses were not included. After correction for these variables, there were no differences between the medical and non-medical costs and, thus, the total healthcare costs in both groups at each time point. Table 2 depicts an overview of the direct medical costs, direct non-medical costs, indirect costs and QALYs.

The UC group had a mean real intervention cost of 219.46 ± 61.79 EUR, while in the I group, a mean cost of 688.70 ± 299.84 EUR was found. Since no between-group differences were observed in mean ectopic fat deposition or mean metabolic parameters, the ACURs were calculated. There was a decrease in the ACUR after six months in both groups. In the UC group, the ACUR decreased from 486.19 EUR to 351.50 EUR. In the I group, the ACUR decreased from 876.23 EUR to 598.76 EUR.

## 4. Discussion

Ectopic fat deposition is more strongly associated with type 2 diabetes and cardiovascular disease risk than generalised obesity [6,7,26]. This study is, to our knowledge, the first one that examined the additional value of an individualised exercise regimen to a hypocaloric diet regarding decreases in ectopic fat deposition in multiple regions.

We cannot conclude that a combined therapy of hypocaloric diet and an exercise prescription is superior to diet intervention alone. The reduction of ectopic fat deposition and amelioration of the metabolic risk was similar in both groups. It is possible that the great changes obtained by a hypocaloric diet masked the additional effects given by exercise therapy [27,28]. Despite the rather small effect of exercise training, which was also concluded in a meta-analysis, there are some RCTs in which high-volume exercise training shows an additional value over diet in the reduction of abdominal and hepatic lipid content [29,30]. In the study by Goodpaster et al., compliance was promoted and controlled and participants could receive small financial incentives for adherence, leading to better exercise compliance and a lower dropout rate (9%). There is good evidence that a higher exercise intensity (in order to achieve a high training volume) yields a greater reduction in ectopic fat and health benefits [31]. Although dose responsiveness between exercise and health effects is well established [32], the question regarding whether unsupervised exercise training is the best manner to achieve a high training volume should be investigated. In this study, half of the fitness centres had a digital registration system in which training intensity, frequency and duration were checked afterwards. It was seen that the prescribed exercise intensity during aerobic training (target heart rate zone) was often not reached during training because participants had manually lowered the working load or training level. Although the training intensity was prescribed based on the results of a personal CPET test, it seemed that patients were not capable of reaching the (rather high) training intensity by themselves. It might be that a more encouraging setting is needed to achieve a greater training volume. In this regard, a controlled and encouraging exercise program in a supervised setting seems to be a better choice in sedentary patients with overweight or obesity. This can lead to a higher training volume with better exercise compliance and a lower dropout rate.

Despite the fact that no between-group differences were found, a significant reduction in ectopic fat of the heart, liver and abdomen was seen after an intervention of six months in a hypocaloric diet, independent of whether it was combined with unsupervised exercise training. A reduction in ectopic fat deposition can yield important clinical benefits, particularly in the liver, where a mean reduction of 3.5% IHL was found. Hepatic steatosis was diagnosed in 17 women at baseline because the IHL content exceeded 5.5% [33]. After the treatment period, the hepatic lipid content was normalised (<5.5%) in 11 of these women. In five women, hepatic lipid content was still higher than 5.5%. Greater reductions in IHL (up to 5%) were found in a meta-analysis of the literature [34].

In this study, no changes in IMCL were found. Previous studies were inconsistent regarding IMCL changes in lifestyle interventions and their relations with metabolic risk [34,35]. A decrease in IMCL is expected following weight loss. Although an increase in metabolic flexibility after exercise training can result in IMCL, which increases together with high oxidative capacity and enhanced insulin sensitivity [36,37,38]. In this regard, it would be interesting if data from IMCL could be compared with insulin action and oxidative capacity.

On the other hand, ectopic fat can also be a flexible fat depot that can increase or decrease after a single bout of exercise or a short-term dietary intervention [39]. Since instructions were given to avoid high-fat-loaded food and exercise training before the clinical exams, we suggest that the obtained results show the effect of the entire treatment.

Besides the assessment of clinical outcomes, an adapted economic analysis was performed. As expected, the healthcare costs observed in this study were skewed [40]. Some participants showed direct medical and non-medical costs that were up to five-fold of the mean costs (from a societal perspective). No significant differences in costs and QALYs between the UC and I groups were found. These results should be interpreted with care, taking the skewness of the data into account.

Because there were no clinical differences in the outcomes between the treatment groups, the ICER was not calculated. The ACUR, which expresses the ratio of the average cost spent per QALY in a single study group (i.e., no incremental analyses between the two study groups), has served as another important measure in cost-effectiveness analyses [41]. In both groups, there was a similar drop in the ACUR. This means that the average cost per QALY decreased, which can be explained by a reduction in societal direct costs after 6 months. The decrease in direct costs was attributed to a reduction in medical costs. Based on the ACUR results, no statement about the cost-effectiveness (i.e., a strategy that costs less and generates more effectiveness) can be made since no incremental analyses between the UC and I groups were performed. The price of the treatment was three times higher in the I group as compared to the UC group. This included the cost of a fitness club membership, patient transport costs and sports gear. Since there was no difference between the results of the UC and I groups, the combination of diet and exercise cannot be defined to be cost-effective for the reduction of ectopic fat deposition. One can argue that this data should be followed by an economic data analysis. Since this would lead to more uncertainties, it was decided to describe the current situation and inspire further research.

Some limitations of our study need to be acknowledged. To begin with, all participants were women, and because of rigorous eligibility criteria, only premenopausal women without diabetes, cardiovascular diseases or thyroid problems were included. Hereby, results might not be generalisable. On the other hand, since the menopausal state was checked, the influence of estrogen levels on fat accumulation was negligible.

Second, the activity of daily living was not controlled, thus the total energy balance was not calculated, which might have influenced the results.

Third, a full economic analysis was not possible since there were no changes in primary and secondary outcomes between the UC and I groups. The use of monthly telephone questionnaires instead of cost diaries could give a more complete overview of costs and may have led to less bias. Given the limited number of participants for economic analysis and the dropout rate of 15%, the results should be interpreted with care.

## 5. Conclusions

In summary, a dietary intervention seemed to be successful at reducing ectopic fat deposition and improving the metabolic risk profile in women with overweight and obesity. The addition of free of charge but unsupervised exercise training was not found to further improve the metabolic health or phenotype over six months.

## Figures and Tables

**Figure 1 ijerph-18-08004-f001:**
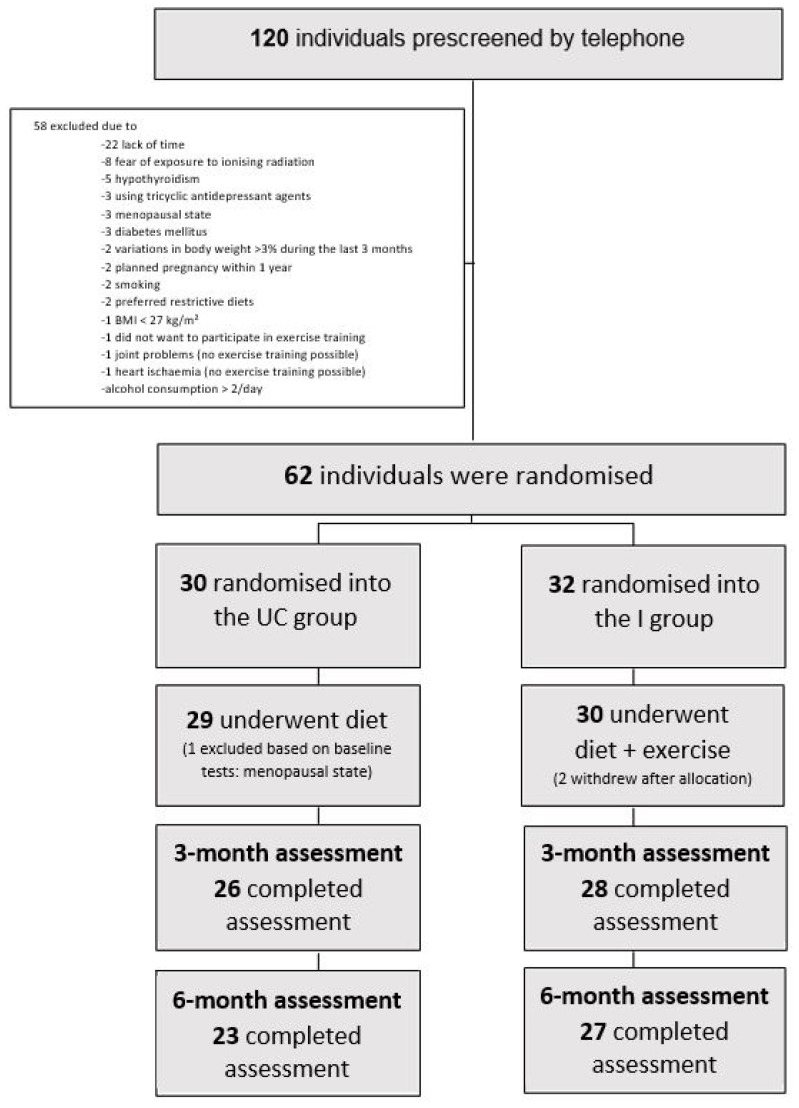
Flow chart of the study participants.

**Figure 2 ijerph-18-08004-f002:**
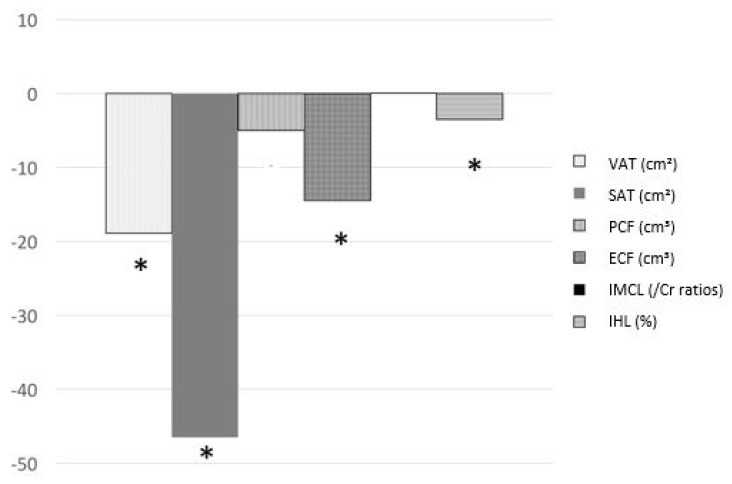
Changes in ectopic fat deposition after a 6-month intervention in the total study sample. VAT-visceral abdominal tissue, SAT-subcutaneous abdominal adipose tissue, PCF-pericardial fat, ECF-epicardial fat, IMCL-intramyocellular lipids, IHL-intrahepatic lipids, * *p* < 0.05.

**Figure 3 ijerph-18-08004-f003:**
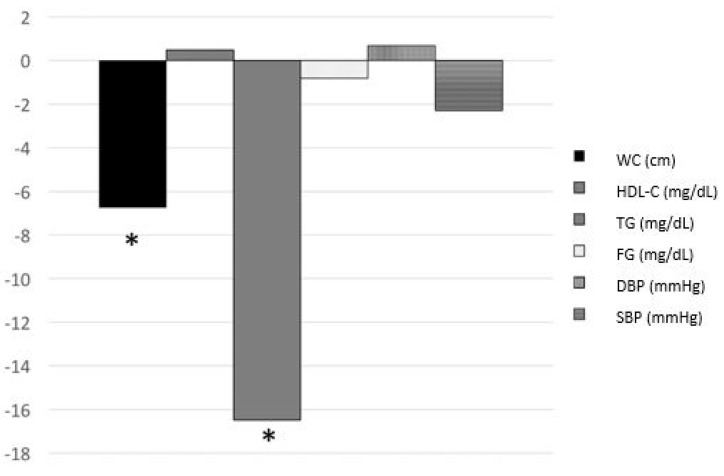
Changes of metabolic parameters after a 6-month intervention in the total study sample. WC-waist circumference, HDL-C-high-density lipoprotein cholesterol, TG-triglycerides, FG-fasting glucose, DBP-diastolic blood pressure, SBP-systolic blood pressure, * *p* < 0.05.

**Figure 4 ijerph-18-08004-f004:**
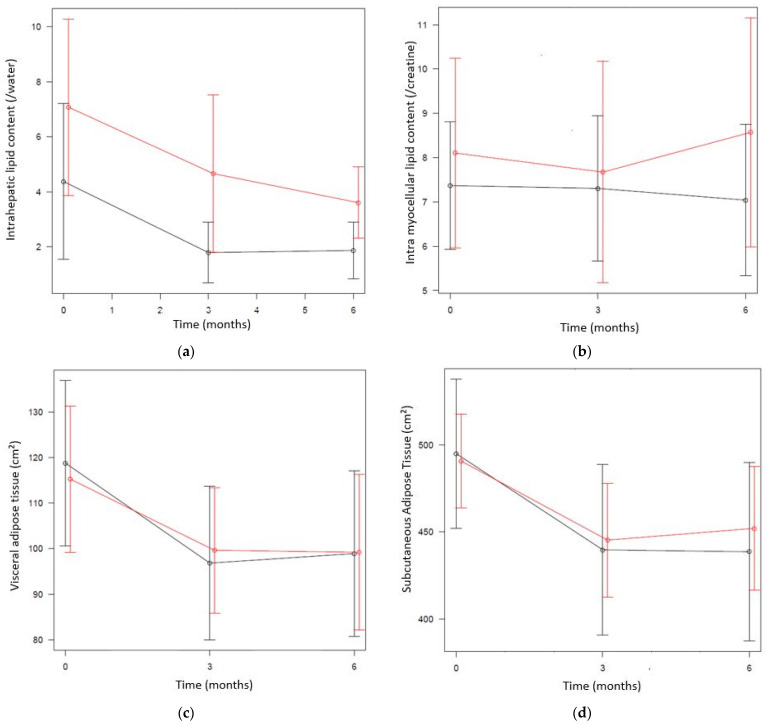
Mean values of ectopic fat deposition and SAT at each time point. The error bars represent the 95% confidence intervals around the mean. Black line: UC group (diet); red line: I group (diet + exercise). (**a**) IHL over time, (**b**) IMCL over time, (**c**) VAT over time, (**d**) SAT over time, (**e**) PCF over time and (**f**) ECF over time.

**Table 1 ijerph-18-08004-t001:** Descriptive statistics of study participants (mean ± SD).

	UC Group: Diet (*n* = 30)	I Group: Diet + Exercise (*n* = 32)	*p*-Value
**General characteristics**			
Age (years)	36.11 ± 8.94	37.64 ± 8.53	0.49
BMI (kg/m^2^)	32.27 ± 3.50	32.98 ± 3.60	0.51
Participants with NAFLD	*n* = 5 (17%)	*n* = 12 (37.5%)	
Participants with metabolic syndrome	*n* = 1 (3%)	*n* = 2 (6.25%)	
**Ectopic fat deposition**			
Intrahepatic lipid content (%)	4.37 ± 7.49	7.08 ± 8.95	0.20
Intramyocellular lipid content (/creatine)	7.37 ± 3.82	8.10 ± 5.99	0.57
VAT (cm^2^)	118.78 ± 48.18	115.27 ± 44.89	0.77
SAT (cm^2^)	493.46 ±112.07	493.87 ± 76.20	0.99
Pericardial fat (cm^3^)	159.47 ± 55.58	150.81 ± 46.94	0.51
Epicardial fat (cm^3^)	53.73 ± 30.32	65.36 ±54.65	0.31
Body fat (%)	39.20 ± 4.21	39.18 ± 3.83	0.98
Fat-free mass (kg)	54.70 ± 5.23	54.11 ± 6.26	0.69
**Metabolic parameters**			
Waist (cm)	100.09 ± 10.13	101.28 ± 8.38	0.61
Triglycerides (mg/dL)	105.22 ± 69.19	99.13 ± 43.19	0.68
Systolic blood pressure (mm Hg)	121.81 ± 14.78	120.00 ± 10.34	0.58
Diastolic blood pressure (mm Hg)	72.48 ±9.94	73.93 ± 9.18	0.55
HDL cholesterol (mg/dL)	57.19 ± 10.72	58.63 ± 11.35	0.55
Fasting glucose (mg/dL)	84.56 ± 7.23	86.30 ± 8.57	0.29

Footnote: *t*-test was used to compare mean values between the UC and I groups with a significance level of 0.05. UC group—Usual Care Group, I Group—Intervention Group, VAT—Visceral Adipose Tissue, SAT -Subcutaneous Adipose Tissue.

**Table 2 ijerph-18-08004-t002:** Change in direct costs and QALY (mean EUR ± SD).

Economic Outcome Parameters	Baseline	*p*-Value (Group)	Month 6	*p*-Value (Group)
**Societal direct medical and non-medical costs**				
UC group	405.48 ± 845.97	0.40	301.94 ± 535.74	0.48
I group	731.65 ± 1669.53		518.53 ± 1359.65	
**Quality-adjusted life year (QALY)**				
UC group	0.83 ± 0.16	0.81	0.86 ± 0.14	0.79
I group	0.84 ± 0.14		0.87 ± 0.13	

Footnote: *t*-test was used to compare mean values between the UC and I groups with a significance level of 0.05. UC group—Usual Care Group, I Group—Intervention Group.

## Data Availability

The cost-of-illness study regarding this population was already published (ref: “Health-related costs in a sample of premenopausal non-diabetic overweight or obese females in Antwerp region: a cost-of-illness analysis”; *Arch Public Health*. 30 July 2018; 76:42 DOI:10.1186/s13690-018-0285-1. eCollection 2018. PMID: 30069308 PMCID: PMC6065060 DOI:10.1186/s13690-018-0285-1).

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
