# Peer review of "Unsupervised Exercise Training Was Not Found to Improve the Metabolic Health or Phenotype over a 6-Month Dietary Intervention: A Randomised Controlled Trial with an Embedded Economic Analysis"

_ijerph, 2021, doi:10.3390/ijerph18158004_

Round 1

Reviewer 1 Report

This submission compared the effectiveness and cost-effectiveness of a hypocaloric diet intervention together with an unsupervised exercise training program with a hypocaloric diet alone to reduce ectopic fat in sixty-one premenopausal women with overweight or obesity. I like to give the following comments.

  1. The ectopic fat did not introduce in clear. Also, for the unsupervised exercise training.
  2. Tissue with attenuation values in the interval of -30 to -190 Hounsfield units considered as fat. It needs reference(s) to support.
  3. The study protocol was described extensively in a previous publication.[10] However, the indicated reference was mainly for magnetic resonance spectroscopy. Please check it.
  4. A higher exercise intensity yields greater reduction of ectopic fat that seems the main factor in the negative data from an unsupervised exercise training program. Please discuss it in detail.
  5. In Table 3, value for errors seems larger than the mean value regarding social costs. Why?
  6. Fasting glucose was not modified by the treatment. Why?
  7. In conclusion, a multidisciplinary lifestyle approach has been recommended. However, it was not described in clear in the discussion.

Author Response

see word doc

Reviewer 2 Report

This manuscript describes results of a randomized control trial comparing numerous health outcomes and costs following 3 and 6 months of hypocaloric diet with (intervention) and without (control) unsupervised exercise training. This is a highly relevant study from a public health perspective that can add to the evidence on what types of strategies are effective (and worth investing in) for weight loss and metabolic health, and in turn the prevention of comorbidities and related health costs. The authors have adequately considered the expected drop-out rate in their calculation of sample size and carried out adequate comparisons of different parameters over time. Concerns related to this manuscript are generally minor: The authors could elaborate more in the result section and table footnotes need to be added. Overall, the authors refer to ectopic fat to address a range of parameters (including VAT and SAT) but also mention “subcutaneous fat” as a broad term (e.g. in Figure 4 footnote). Other terms used broadly were “fat parameters”, “metabolic parameters”, “metabolic syndrome parameters” “health parameters” and “phenotype”, encompassing the measurements performed. It would be useful to the reader if the authors were more consistent with wording throughout the text to avoid confusion. Please consider other more specific comments given:  

Line 49: “…and an increased risk of nearly every chronic condition”. Many inherited, autoimmune, or viral chronic conditions would not necessarily be related to obesity. The use of .”nearly every” seems like an overstatement, even if cited from other authors.  

Line 149: Perhaps a short description of the cost diary used (e.g. what information was requested; what instructions were given) would be helpful to the reader.

Line 210-12: The template instructions need to be removed.

Line 224: The authors refer to Table 1, but this table is labelled Table 2. Additionally, this descriptive table could do with a footnote indicating the test used to compare groups, and the alpha level for significance.  

Figure 2: Footnote: VAT and SAT are referred to under the umbrella of ectopic fat parameters. Is this the accurate terminology? Same in line 244 and elsewhere.

Line 235-38: For some results the F(df1,df2)=F-ratio is given in the text, but not for others. Please add for consistency.

Table 2: Should be Table 1. Also, heading “Change” is misleading. The values are presented for each timepoint (not the change over time). Please indicate that the values on the left 3 columns are means and the values on the right are p-values. A footnote is also missing, where the applied test and the significance level for p-values are indicated.  

Table 2 results: a short description of the significant changes observed for specific parameters could be added in the text. Of note: the table indicates a significant reduction of fat free mass in both groups. This reduction would inherently affect body fat %, leading to a greater % than if fat free mass were higher. Thus body fat reported as % may be somewhat misleading. The authors might consider fat mass index (kg/m2) and fat free mass index instead, which are independent of eachother.  

Line 318: “It was seen that the prescribed exercise intensity during aerobic training (described with a target heart rate zone) was often not reached during training because participants had lowered the training intensity.” Is it possible that there was a gain in fitness over time during which the intensity of the exercise was not necessarily reduced, but rather the participants became fitter and hence the same intensity of exercise was not sufficient to reach the target heart rate?

Line 361: specify – unsupervised exercise (and possibly mention – constant intensity rather than increasing intensity if this was the case).

Line 370: “Since the limited amount of participants for economic analysis and the dropout rate of 15%, results should be interpreted with care”. Change Since to Given….

Furthermore, it seems from this sentence that there were fewer participants included in the economic analysis than in the analysis of health outcomes. Is this the case. If this was the case, it needs to be made clear earlier in the manuscript.

English and punctuation corrections:

Line 34: The following sentence needs revising: “This goal can be reached by different strategies going from non-invasive forms of behaviour therapy, pharmacotherapy or invasive approaches weight loss surgery.”

Line 96: “…without taken possible…” should be “without taking possible”

Line 104: “times/week. (supplementary file; appendix 1- training scheme)”. Please check punctuation.

Line 124: “…according previously...”. Should be “according to previously…”

Line 147: Full stop is missing.

Line 265: “Additional, The indirect costs”. Please revise typos.

Line 307: “Despite the rather small effect of exercise training was also concluded in the meta-analysis of the literature…”. Please revise English in this sentence.

Author Response

see word doc

Reviewer 3 Report

This is an interesting RCT that concluded that unsupervised exercise training cannot improve the metabolic health or phenotype over a 6-month dietary intervention.

  1. Please revise the affiliations to include country etc.
  2. Why was the cut-off of 27 kg/m2 selected for the BMI?
  3. The in-line citing of references should be revised, the dot (.) should follow the brackets [], e.g., [1]. not .[1] etc.
  4. Please delete lines 210-212.
  5. line 385: please correct/update.
  6. Were any of the postmenopausal women taking hormone replacement therapy? It has been shown that these drugs impact on the lipid profile of females. Could they also act on ectopic fat? Please see: https://pubmed.ncbi.nlm.nih.gov/33617974/ & https://pubmed.ncbi.nlm.nih.gov/33865986/
  7. Were any of the women involved in the RCT taking (herbal) supplements? Several natural compouns have been shown to have anti-obesity effects.  Maybe in the future the investigators can check the impact of supplements + diet + physical exercise. See:

https://pubmed.ncbi.nlm.nih.gov/32372444/

https://pubmed.ncbi.nlm.nih.gov/33922341/ 

Overall, a well-written paper suitable for publication following these minor revisions.

Author Response

see word doc
